# The Expression of miRNAs Involved in Long-Term Memory Formation in the CNS of the Mollusk *Helix lucorum*

**DOI:** 10.3390/ijms24010301

**Published:** 2022-12-24

**Authors:** Gennady V. Vasiliev, Vladimir Y. Ovchinnikov, Pavel D. Lisachev, Natalia P. Bondar, Larisa N. Grinkevich

**Affiliations:** 1The Federal Research Center Institute of Cytology and Genetics, Siberian Branch, Russian Academy of Sciences, 10 Lavrentiev Avenue, Novosibirsk 630090, Russia; 2Federal Research Center for Information and Computational Technologies, 6 Lavrentiev Avenue, Novosibirsk 630090, Russia; 3The Federal State Budget Scientific Institution Pavlov Institute of Physiology, Russian Academy of Sciences, 6 nab. Makarova, St. Petersburg 199034, Russia

**Keywords:** *Helix lucorum*, CNS, microRNA, next generation sequencing, long-term memory, MIR-10 family, 5,7-DHT

## Abstract

Mollusks are unique animals with a relatively simple central nervous system (CNS) containing giant neurons with identified functions. With such simple CNS, mollusks yet display sufficiently complex behavior, thus ideal for various studies of behavioral processes, including long-term memory (LTM) formation. For our research, we use the formation of the fear avoidance reflex in the terrestrial mollusk *Helix lucorum* as a learning model. We have shown previously that LTM formation in *Helix* requires epigenetic modifications of histones leading to both activation and inactivation of the specific genes. It is known that microRNAs (miRNAs) negatively regulate the expression of genes; however, the role of miRNAs in behavioral regulation has been poorly investigated. Currently, there is no miRNAs sequencing data being published on *Helix lucorum*, which makes it impossible to investigate the role of miRNAs in the memory formation of this mollusk. In this study, we have performed sequencing and comparative bioinformatics analysis of the miRNAs from the CNS of *Helix lucorum*. We have identified 95 different microRNAs, including microRNAs belonging to the MIR-9, MIR-10, MIR-22, MIR-124, MIR-137, and MIR-153 families, known to be involved in various CNS processes of vertebrates and other species, particularly, in the fear behavior and LTM. We have shown that in the CNS of *Helix lucorum* MIR-10 family (26 miRNAs) is the most representative one, including Hlu-Mir-10-S5-5p and Hlu-Mir-10-S9-5p as top hits. Moreover, we have shown the involvement of the MIR-10 family in LTM formation in *Helix*. The expression of 17 representatives of MIR-10 differentially changes during different periods of LTM consolidation in the CNS of *Helix*. In addition, using comparative analysis of microRNA expression upon learning in normal snails and snails with deficient learning abilities with dysfunction of the serotonergic system, we identified a number of microRNAs from several families, including MIR-10, which expression changes only in normal animals. The obtained data can be used for further fundamental and applied behavioral research.

## 1. Introduction

One of the most challenging tasks of fundamental neurobiology is to elucidate the role of epigenetic mechanisms of brain function, including neuronal differentiation, diverse behavior repertoires, responses to stress, and formation of long-term memory (LTM). Epigenetic processes, such as DNA methylation and post-translational modifications of histones lead to significant changes in the spatial structure of chromatin and consequently changes in the expression of target genes [1]. Indeed, regulation of gene expression has been found also to occur via RNA interference involving microRNAs (miRNAs) [2]. Moreover, it has been shown that the regulation of histone modifications and miRNAs can be coordinated to control gene expression [3]. MiRNAs are a family of small well-conserved single-stranded RNAs, about 20 nucleotides in length, which repress gene expression by promoting mRNA degradation or by inhibiting protein translation [4,5]. The biogenesis of miRNAs depends on a few evolutionary conserved proteins: DROSHA, DGCR8, EXP5, RAN, DICER, TARBP2, AGO, and PIWI [6]. Involvement of miRNAs in neuronal differentiation [2,7] and in LTM formation in both vertebrates and invertebrates (rats, mice, zebrafish, mollusks, drosophila) has been demonstrated [8,9,10,11,12,13,14,15,16]. In addition, there is increasing evidence of the involvement of miRNAs in the pathogenesis of diseases associated with mental disorders including Alzheimer’s disease (AD), schizophrenia, and senile dementia [17,18,19,20,21]. These miRNAs are expected to be used as targets for therapeutic treatment [22]. Challenges of research in this area are related to the fact that the central nervous system (CNS) of an animal consists of a multitude of cells with various functions and with different miRNA pools being expressed. While about one hundred miRNAs can be functional, each of them may have several mRNA targets [23]. Indeed, a specific mRNA can be targeted by several miRNAs [24]. Considering such a high level of complexity, it is understandable why the functional role of most miRNAs has not been defined yet. Thus, the use of animals with simpler nervous systems in studying the miRNA function is a promising research tool.

Mollusks (such as *Aplysia*, *Helix*, and *Lymnaea*) are popular objects for studying the molecular mechanisms of neuroplasticity [25,26,27,28]. The use of the model of long-term facilitation of the sensorimotor synapse in *Aplysia* allowed Eric Kandel and co-authors to discover the basic mechanisms of plasticity [25], for which E. Kandel was awarded the Nobel Prize. They were also the first to describe the important role of microRNA miR-124 in the mechanisms of plasticity in neurons in the cell culture, but no animal studies were performed [8].

In our lab, we use terrestrial mollusk *Helix* to investigate molecular mechanisms of memory formation. Mollusks display a rich behavioral repertoire while having relatively simple CNS comprising giant neurons. The advanced olfactory system makes the animal successful in finding food. Passive defensive behavior is the dominant reaction in response to dangerous stimuli, while the animal quickly hides in the sink. In this regard, *Helix* can be trained for several conditioned defensive reflexes. Our investigations are focused on the conditioned reflex of food aversion, which is formed in response to the association of food with a pain stimulus and is a good model for studying the molecular mechanisms of long-term memory [29,30,31,32]. This reflex is well investigated at the behavioral level, and the neural network underlying it has been identified [33]. An important role in the formation of defensive reflexes in mollusks, including *Helix*, is played by serotonin, which mediates the action of the nociceptive unconditioned stimulus [25,33,34]. Maturation of the serotoninergic neural system in *Helix* correlates with the appearance of long-term behavioral and synaptic sensitization, and with the appearance of food-avoidance conditioning abilities [29,32,35,36]. Suppression of the serotoninergic system leads to a deterioration in the formation of long-term forms of conditioned defensive reflexes [29,36,37]. We have shown that several transcription factors (TF) and epigenetic modifications of histones are involved in the conditioned aversive reflex formation and these processes are induced by serotonin through activation of the MAPK/ERK regulatory cascade [30,31,32,38,39]. These epigenetic modifications are under the control of both the activation and inhibition pathways involved in the formation of LTM. However, the involvement of microRNAs in the formation of long-term memory in *Helix* has not been studied. Difficulties are associated with the fact that the genome of this animal has not been sequenced, and miRNAs have not been sequenced, either.

In this regard, we have performed sequencing followed by comparative bioinformatics analysis of the miRNAs from the CNS of *H. lucorum* to enable further investigations in this important area of neuroscience. We have identified 95 distinct miRNAs, many of which are involved in different CNS processes of vertebrates, including LTM formation. In addition, we have shown that the MIR-10 family (26 miRNAs) is the most representative, and some of its members are involved in the LTM formation. Our comparative analysis of microRNA expression in normal animals and in animals with dysfunction of the serotoninergic system, in which the formation of long-term conditioned defensive reflexes is impaired, made it possible to identify the miRNAs involved in the formation of this reflex. This work provides an important background for future studies of the role of miRNAs in fear-dependent memory formation.

## 2. Results

### 2.1. Identification of miRNAs Expressed in the CNS of Helix

To identify miRNAs from the CNS of *Helix lucorum*, we performed sequencing analysis of *H. lucorum* small RNAs isolated from the CNS of 4 animals using miRNA-seq followed by bioinformatics analysis. The total number of unprocessed reads in every sample was 13 million (13,477,340), 12 million (12,195,228), 15 million (15,153,866), and 14 million (14,675,100); after the filtering process, the final number of reads was 8 million (8,727,391), 7 million (7,206,552), 9 million (9,297,789), and 8 million (8,671,088), respectively (representative samples). The raw reads were submitted to the NCBI Sequence Read Archive (SRA) under the accession number SRP136226.

We identified 95 candidates of conserved miRNAs belonging to the families of miRNAs described in the mollusks *Lottia gigantea* and *Crassostrea gigas* (MirGeneDB). The sequences of all the identified miRNAs in the *H. lucorum* CNS are presented in Appendix A.

In this paper, we have used our own identifiers (IDs) designed similarly to MirGeneDB IDs. Yet, instead of ‘P’ denoting the paralogue, we used ‘S’ denoting the sequence, since there is no genome assembly for *H. lucorum*; so, we could not compare miRNA genes and accurately indicate the number of the paralogs. We preferred to use MirGeneDB IDs instead of miRBase IDs, as the miRBase nomenclature has problems described in [40]. Here we showed miRBase ID in parentheses at the first mention of the miRNA gene.

The top ten miRNAs expressed in the *H. lucorum* CNS are shown in Figure 1. Two of which Hlu-Mir-22-S1_3p and Hlu-Mir-153-S1_3p (highlighted in black) are homologues of Mir-22-P1 (Mir-22) and Mir-153-P1/P2 (Mir-153) genes in other species correspondingly, which play a significant role in the functioning of the CNS in vertebrates and the marine mollusk *Aplysia californica* [41,42,43].

Hlu-Mir-10-S5_5p is miRNA with the highest expression level in the *H. lucorum* CNS. Our analysis showed this miRNA to be homologous to Mir-10-P2d (Mir-100), Mir-10-P2b (Mir-99b), and Mir-10-P2c (Mir-99a) mature sequences of *Homo sapiens* and corresponding genes of *Lottia gigantea*, *Crassostrea gigas*, and *Aplysia californica* (Figure 2, Appendix A).

All miRNAs shown in Figure 2 have the same seed region (the region between the 2nd and 8th nucleotides), which is necessary for the recognition of the target mRNA [40]. The seed region is the most conserved part of mature miRNAs. Another important region is the site from 13th to 16th nucleotide that increases the efficiency of target mRNA recognition. This site in Hlu-Mir-10-S5_5p is identical to the human site and to those shown in the figure of mollusk miRNAs. Thus, Hlu-Mir-10-S5_5p would be very interesting for our further research.

It Is worth noting that the bioinformatic analysis of miRNA in *H. lucorum* revealed another interesting feature of the MIR-10 family: it is an unusually large number of miRNAs belonging to that family (Appendix A). According to our data, *H. lucorum* has 26 miRNAs from the MIR-10 family, while the *L. gigantea* and *C. gigas* have only 5 miRNAs from this family (according to the data from MirGeneDB). Perhaps this diversity is a result of the various representation of MIR-10 members in different tissues. In *L. gigantea* and *C. gigas*, miRNAs were isolated from the whole animal and haemocytes respectively (the data from MirGeneDB 2.1 database). The number of genes expressed in the CNS is much higher than in any other organ or tissue [44], and, accordingly, more complex regulation of their expression is required and, hence, more miRNAs are expressed in the CNS. All the MIR-10 family sequences from *H. lucorum* can be divided into three subfamilies: MIR-10-P1 (Mir-10); MIR-10-P2 (Mir-99/100), and MIR-10-P3 (Mir-125) according to seed sequences. After alignment, Hlu-Mir-10-S1/S2/S3 were assigned to the subfamily MIR-10-P1; Hlu-Mir-10-S4/S5/S6/S7—to the subfamily MIR-10-P2. The remaining sequences were referred to the subfamily MIR-10-P3 (Table 1).

Among *Helix* miRNAs, we also identified miRNAs, belonging to MIR-22, MIR-33, MIR-124, MIR-137, and MIR-153 families (Appendix A), which are known to be involved in different processes of the vertebrate CNS, including fear behavior and LTM [43,45,46,47,48,49]. The data of our comparative analysis regarding the homology of Hlu-Mir-9_5p, Hlu-Mir-124_3p, and Hlu-Mir-153-S1/S2_3p to mature miRNAs from *H. sapiens*, L. *gigantea*, *C. gigas,* and *A. californica* is presented in Figure 3.

### 2.2. miRNAs Expression Dynamics in the Helix CNS after Learning

Further, in order to study miRNAs involved in the formation of LTM we analyzed miRNA expression in the *Helix* CNS at different time points after the food-avoidance conditioning using miRNA-seq and bioinformatics analysis. The analysis of 19 samples showed that the level of 34 miRNAs in the *Helix* CNS increased or decreased differentially 2 and/or 5 h after training (Figure 4A,B, Appendix A). The number of miRNAs whose expression changed 5 h after training was much greater than after 2 h. Interestingly, 5 h after training, out of 13 miRNAs with increased expression, 11 belong to the MIR-10 family, while among 18 miRNAs with decreased expression, there was only one representative of this family (Mir-10-S15).

Two h after training one MIR-10 family miRNA (Mir-10-S6) decreased and none of the MIR-10 family microRNAs increased. In addition, 2 *h* after training Mir-76-S3, -133, and -1985-S2_3p expression was induced and Mir-2722-S1_3p expression was reduced (Figure 4A, Appendix A). Five h after training, besides MIR-10, 21 miRNAs undergo differential expression, among them Mir-92-S1/S2, -133, -124, -137, and -153-S1/S2 (Figure 4B), belonging to the families which are widely studied in connection with cognitive processes in vertebrates and other organisms. The expression of Mir-133 and Mir-2722_3p changed both 2 and 5 h after training, but in opposite directions at different time points (Figure 4A,B, highlighted by hatched columns).

To identify microRNAs specifically associated with long-term memory formation, we compared the level of miRNA expression in the CNS of normal animals and animals with learning deficits subjected to training.

### 2.3. miRNAs Expression in the CNS of Learning Deficient DHT-Treated Helix after Training

As noted above, the neurotransmitter serotonin plays an important role in the formation of conditioned avoidance reflexes in Helix. Dysfunction of serotoninergic terminals caused by the injection of neurotoxin 5,7-dihydroxytriptamine (DHT) reduces the ability of animals to form these paradigms of plasticity [29,37,50]. Thus, we analyzed miRNA expression in animals injected with DHT using miRNA-seq and bioinformatics analysis. DHT was injected at a dose of 20 μg/g of animal weight in two draughts at six day intervals. Learning experiments were carried out seven days after the last injection using the same training protocol as for DHT untreated animals. It was shown that in naive animals, only four miRNAs demonstrated a significant change in expression upon administration of DHT, namely Mir-124, -317, -10-S13, and -1992 (Appendix A). However, the expression of a significant number of miRNAs changed in trained DHT-treated animals (Figure 5).

Eleven miRNAs were found to be differentially expressed 2 h post-training in DHT-treated animals, of which five miRNAs were down-regulated and six up-regulated (Figure 5A). Five hours post-training, the number of differentially expressed miRNAs increased significantly, reaching 29 (Figure 5B). Again, both down-regulated and up-regulated miRNAs were represented significantly. Among microRNAs whose expression increased 5 h after training, the majority of miRNAs belong to the MIR-10 family (eight species), and only one MIR-10 family miRNA (Mir-10-S15) showed a decrease. Profiles of differentially expressed miRNAs at the studied time points differed; however, three miRNAs showed a similar expression behavior, namely Mir-2-S1_3p, Mir-277, and Mir-1992 (Figure 5A,B, highlighted by hatched columns). However, it is not clear whether the changes in the expression of these three miRNAs were long-lasting or developed during separate waves of gene expression [37].

Further analysis showed that the injection of DHT significantly changed the set of miRNAs differentially expressed as a result of training, compared with animals that did not receive DHT (Table 2).

Two hours after training, expression of only one MIR-10 family member decreased in DHT-untreated animals (Mir-10-S6), while in DHT-treated animals, expression of three other MIR-10 family miRNAs (Mir-10-S12, -10-S18, and -10-S19) decreased and expression of Mir-10-S5 increased (Table 2). In addition, DHT-treated animals differentially expressed five microRNAs from other families, namely Mir-2-S1_3p, -33, -193, -277, and -278. This is in contrast to DHT-untreated animals, where the expression of four other microRNAs changed (Mir-76-S3, -133, -1985-S2_3p, and -2722-S1_3p).

A different expression pattern was observed 5 h after training. Eight MIR-10 family miRNAs demonstrated unidirectional dynamics in DHT-treated and DHT-untreated animals (Table 2), while the dynamics of four MIR-10 family miRNAs were different. In DHT-untreated animals, the expression of Mir-10-S21, -10-S24, and -10-S25 increased, while in DHT-treated animals, their expression did not change. Moreover, none of the MIR-10 family miRNAs changed both 2 and 5 h after training. Five hours after training, there was a decrease in the expression of one microRNA and an increase in the expression of 11 representatives of the MIR-10 family (all of them belong to the Mir-125 subfamily).

As for the expression of microRNAs of other families, also in some cases, the expression changed in the same direction in DHT-treated and DHT-untreated animals, while in other cases differential expression was observed in one animal group, and the absence of significant changes in the other animal group (Table 2). Twelve microRNAs demonstrated unidirectional expression changes (11 miRNAs decreased and one increased), while 15 microRNAs showed differences in dynamics (of a type specified above) in DHT-treated and DHT-untreated animals.

In general, an exceptionally large representation of MIR-10 family members among all microRNAs differentially expressed in the CNS at different time intervals after the training should be noted. Moreover, among them, there were microRNAs belonging to different types, differing for their potential targets (Table 1). Therefore, we selected three representatives of the MIR-10 family, the most highly expressed in the *Helix* CNS, to confirm the results of sequencing by PCR. Those were Mir-10-S5 (Mir-100), Mir-10-S9, and Mir-10-S10 (Mir-125). As can be seen in Figure 6, PCR analysis confirmed the different dynamics of the Mir-100 and Mir-125 subfamilies miRNAs after training. Specifically, the expression of Mir-10-S5 (Mir-100) did not change after learning in normal animals, but increased after training in DHT-treated animals, while the expression of Mir-10-S9 and Mir-10-S10 (Mir-125) did not differ in normal animals and DHT-treated animals.

Thus, a comparison of the expression of miRNAs isolated from the CNS of normal animals and animals with dysfunction of the serotonergic system subjected to the training procedure made it possible to detect three groups of miRNAs whose expression changes upon training: miRNAs which expression changes only in animals with native serotonergic system; microRNAs which expression changes only in animals with impaired serotonergic system; microRNAs which expression changes in the same direction in both groups of animals (Table 2). A significant number of miRNAs differentially expressed during learning belongs to the MIR-10 family.

## 3. Discussion

Sequencing and comparative bioinformatics analysis of the miRNAs from the CNS of *Helix* allowed us to detect 95 distinct miRNAs, including miRNAs belonging to MIR-9, MIR-10, MIR-29, MIR-92, MIR-124, MIR-133, MIR-137, and MIR-153 families. The above microRNAs are known to be involved in different processes in the vertebrate CNS, including pain regulation, fear behavior, development of the olfactory system and LTM [14,42,43,46,47,49,51,52]. The MIR-10 family, numbering up to 26 members, is most represented in the *Helix* CNS. In the modern classification of MirGeneDB, the MIR-10 family includes subfamilies directly Mir-10, Mir-99/100, and Mir-125 (Table 1). In vertebrates, all the above-mentioned MIR-10 subfamilies are to some extent involved in neuroplasticity [14,53,54,55,56,57,58]. In particular, Mir-10 and its target genes may regulate development of the nervous system [59].

The main member of the Mir-99/100 group in the *Helix* CNS is Hlu-Mir-10-S5, which has significant similarity to mammalian Mir-100 and is the top expressed among all miRNAs in the CNS of *Helix* (Figure 1). Hlu-Mir-10-S9 is the second highly expressed miRNA in the Helix CNS after Hlu-Mir-10-S5 (Figure 1) and belongs to the Mir-125 subfamily, numbering up to 19 members in Helix (Table 1). Unfortunately, the *Helix* genome has not been sequenced, which complicates analysis of the causes for the diversity of microRNAs. In the case of the Mir-125 subfamily in *Helix*, gene amplification [60] seems most likely, as the members of the family have an almost identical sequence. None of the family members can be derived from another by 3’-terminal tailing or trimming [61], as well as due to imprecise cleavage by Dicer [62], since almost all of these miRNAs are 22 nt long, and two exceptions (21 nt) are associated with deletions of “internal” nucleotides (Appendix A).

As for the possible functional significance of Mir-125 diversity in the *Helix* CNS, it should be noted that even small changes in miRNA sequences outside the seed region can alter the target preference of this molecule, and each miRNA can regulate a large number of genes with a cooperative function in the regulatory networks [63]. The key to solve this issue can be in the dataset on tissue-, sex-, and age-specific expression of 43 MIR-184 genes in *C. gigas* [64]. Interestingly, *C. gigas* has only one dominant Mir-184 gene, which is constantly highly expressed in a large amount cross the whole body, while the rest are tissue- and age-specific. Among Mir-125 expressed in the CNS of adult *H. lucorum*, we also found one variant (Hlu-Mir-10-S9), which is hugely expressed (more than 25% of all microRNAs), and a couple of variants (Hlu-Mir-10-S10 and -S17) which are moderately expressed (400–2000 rpm), while the expression of others varies between 3 and 60 rpm. Therefore, by analogy with the MIR-184 family in *C. gigas*, it can be assumed that snail Mir-125 variants, minor for the CNS of adult snails, play a more prominent role in other tissues and/or at early stages of brain development. It is shown that Mir-125 is associated with the regulation of neuronal progenitors’ differentiation and synaptic transmission [54,55,56,65,66,67].

In addition, the CNS is composed of heterogeneous cell populations that have different transcriptomes and may require different microRNAomes to regulate them. Moreover, the neuronal genome of mollusks is characterized by very high polyploidy. The relationship of increased representation of the MIR-10 family of miRNAs to polyploidy is very probable. Thus, the diversity of miRNAs belonging to the MIR-10 family can reflect their significant role in the most complicated processes occurring in the *H. lucorum* CNS. Thus, in order to elucidate why the evolution of snails of the genus *Helix* led to stabilization of the diversity of the MIR-10 family miRNAs, it is necessary to study the cellular and age specificity of transcriptomes in the future. Further, we showed the MIR-10 family (13 representatives) are differentially expressed in the CNS of *Helix* at different periods of long-term memory consolidation. The damage of serotoninergic transmission by DHT impaired the learning and changed profiles of microRNAs differentially expressed after training (Table 2).

In normal animals, 2 h after training, there was a decrease in the expression of Hlu-Mir-10-S6 (subfamily Mir-99/100), and 5 h after training, the expression of Hlu-Mir-10-S21, -10-S24, -10-S25 (subfamily Mir-125) increased (Table 1, Table 2). It is important that the expression of these miRNAs did not change after training in learning-deficient DHT-treated animals, which suggests their potential involvement in the development of food-avoidance conditioning. Four MIR-10 family miRNAs differentially expressed after training only in DHT-treated animals (Table 2), these are Mir-10-S5 (subfamily Mir-100) and three members of the subfamily Mir-125. These miRNAs might be involved in compensatory molecular mechanisms triggered in response to dysfunction of the serotonergic system. However, most miRNAs of the MIR-10 family differentially expressed after training showed similar dynamics both in DHT-treated animals and in animals not treated with DHT, which suggest their involvement in general molecular mechanisms associated with the response to training. All these miRNAs belong to the subfamily Mir-125.

The MIR-10 family is known to play an important role in the functioning of the CNS in vertebrates and others species. For example, Mir-10a and Mir-10b control the level of the brain-derived neurotrophic factor BDNF [57,68] required for LTM formation [53,57,58,69]. The Mir-10b-dependent decrease in BDNF expression is accompanied by significant cognitive impairment, causes neuronal dysfunction and death, including the case of Huntington’s disease [69]. In addition, it has been shown that formation of LTM (passive-avoidance memory) is associated with reduced expression of Mir-10 and an increase in p-CREB/CREB, C-FOS, and BDNF in the PFC and hippocampus. Stress causes deterioration of this type of LTM through the induction of Mir-10 expression and a decrease in the level of p-CREB/CREB, C-FOS, and BDNF [14]. Mir-10a is involved in the regulation of pain [70]. As noted above, snails have a highly developed defensive behavior, which is implemented in response to dangerous stimuli including pain; thus, the high expression level of MIR-10 in *Helix* CNS can reflect the involvement of these miRNAs in the processes connected to pain, allowing the animals to cope with it and to avoid it.

Mammalian Mir-100 is associated with Alzheimer’s disease, which suggests the possibility of its use for diagnostic purposes, since Mir-100 is found in patients in peripheral blood [20]. It has been recently shown that miR-100 can act as extracellular signaling molecule and participate in the triggering of neurodegeneration [21]. The Mir-125 subfamily is also widely involved in plastic rearrangements of the CNS, and disruption of their expression is associated with Alzheimer’s disease and Fragile-X syndrome (reviewed in [66]). These miRNAs differ in targets associated with synaptic function. Mir-125a targets include PSD-95 [55], Tau [71], and a subunit of the glutamate receptor GluN2A [54]. In honeybees, Mir-100 and Mir-125 are involved in functional specialization (whether a bee will be the queen or a worker honeybee), and this choice is determined by nutrition [72].

The second represented miRNA family in the *Helix* CNS is MIR-92 (5 miRNAs) (Appendix A). Moreover, Mir-92-S1 is among the ten most highly expressed miRNAs (Figure 1). Three representatives of this family (Mir-92-S1, -92-S2, and -92-S3) are differentially expressed in the *Helix* CNS after training (Table 2). MIR-92 is known to play an important role in the CNS, with connections to chronic stress and depression in vertebrates [51] and is necessary for formation of the contextual fear memory [73]. The Mir-17-92 cluster regulates anxiety [74] and is involved in neural progenitors’ differentiation (reviewed in [66]). Confirmed Mir-92a targets include the glutamate receptor subunit GluA1 mRNA [75]. Mir-92 is involved in age-related behavioral changes in honeybees [76].

We have also detected several microRNAs in the *Helix* CNS, including Mir-22, -29, -124, -137, and -153 the function of which is widely studied in connection with cognitive processes in different animal species [43,46,47,48,49,77]. Moreover, the miRNAs of these families undergo differential expression after the food aversion training in *Helix* (Table 2). Present among them is Mir-153-S1, which is abundant in the *Helix* CNS (Figure 1). These miRNAs are highly conserved among mollusks and humans (Figure 3). Moreover, a decrease in the expression of Mir-153-S1 and Mir-153-S2 after training was observed in the CNS in well-learning *Helix* but not in learning-deficient DHT-treated animals. A similar pattern of expression is also observed for Mir-22. Thus, our findings indicate the involvement of Mir-153 and MiR-22 in memory formation of *Helix* and are of interest for further in-depth research.

Mir-153 is involved in the LTM in vertebrates [43,49,52]. So, Mir-153 expression is induced in the hippocampus after contextual fear conditioning [43]. Mir-153 dysregulation is associated with a decrease in the ability for learning and memory formation through impaired BDNF expression in ‘autistic’ mice [49]. Mir-22 is involved in spatial memory impairment in Alzheimer’s disease model in mice [48], and overexpression of Mir-22 protects synaptic structures from degradation and inhibits neuronal apoptosis [78]. Mir-22 was found in the mollusk *Aplysia*, where it gates long-term heterosynaptic plasticity through presynaptic regulation of a functional prion CPEB at multiple binding sites on the mRNA of CPEB and inhibits it in the basal state [42]. CPEB plays an important role in the mechanisms of plasticity in both vertebrates and invertebrates [25]. In contrast to the mouse AD model, the improvement in the efficiency of synaptic transmission in *Aplysia* is caused by downregulation of Mir-22 through the serotonin-induced MAPK/ERK signaling cascade [42]. We also observed a decrease in Mir-22 expression after training in *Helix*. In addition, we have previously demonstrated involvement of serotonin-induced MAPK/ERK signaling cascade in long-term memory formation in *Helix* [29].

In the *Helix* CNS, we observed a decrease in Mir-124 and Mir-137 expression after training in both DHT-untreated animals and DHT-treated animals, which seems to indicate their involvement in the basic molecular processes associated with training. Mir-124 is involved in the formation of long-term synaptic facilitation induced by serotonin in the sensorimotor synapse of *Aplysia*, where Mir-124 regulates the expression of the transcription factor CREB1 [8]. This work also discusses the conservation of the structure and functions of miRNAs in mollusks and mammals, including direct repression of the CREB1 through Mir-124. The above is also supported by the work [79] in which, through blocking the function of Mir-124 using anti-miRNA oligonucleotides has been shown an important role of Mir-124 in memory formation in honeybees. In mice, Mir-124 is involved in the mechanisms of spatial learning and social interactions [80]. In this case, the target of Mir-124 is the transcription factor Zif268. Inhibition of Mir-124 in vertebrates improves memory, potentially through altered levels of genes associated with synaptic plasticity [47]. Additionally, Mir-124 participates in the mechanisms of neuropathic morphine-induced pain and inflammatory muscle pain [71,81]. There are ongoing discussions on the clinical potential of these miRNAs.

Mir-137 plays an important role in the learning in various animal species. Dysregulation of Mir-137 leads to memory deficits, including those observed in autism and schizophrenia, which indicates the great importance of Mir-137for functioning of the human brain (reviewed in [66,82,83,84]). Mir-137 is involved in the regulation of pre- and postsynaptic signaling [85] and has neurogenic properties (reviewed in [66]). This miRNA participates in hippocampus-dependent LTM induced by fear, its overexpression impairs contextual fear memory [46]. Mir-137 was identified in the CNS of the pond snail *Lymnaea*. It has been demonstrated that Lym-Mir-137 is required for single-trial induced LTM (the reflex with positive reinforcement) and its target is the transcription factor CREB2, which inhibits gene expression [27]. CREB2 is involved in the formation of LTM in many animal species, including mollusks [25]. Taking into account the above functions of Mir-124 and Mir-137, as well as the fact that a painful stimulus was used during training, it can be assumed that these miRNAs, the expression of which changed in a similar way in well-learning and learning-deficient (DHT-treated) animals, are involved in pain reaction. In addition, it is possible that the expression of Mir-124 and Mir-137 in DHT-treated and DHT-untreated Helix may differ in other time intervals.

In DHT-treated *Helix* with dysfunction of the serotonin system, we observed changes in the Mir-29 level after training (Table 2). The miRNAs of the MIR-29 family are involved in regulation of dendritic spine morphology [86] and in control of behavior [5] in vertebrates. Mir-29b is downregulated in hippocampus during fear memory formation, which leads to changes in the DNA methylation and changes in the expression of the genes involved in plasticity [77], and Mir-29a regulates the serotonin receptor 5-HT7 expression [87]. As noted above, serotonin plays an important role in the formation of the food aversion reflex in *Helix* [30,32,38]. In addition, Mir-29 is able to modulate synapse morphology and memory processes via the p38 MAPK signaling pathway [88]. This regulatory pathway is also involved in the formation of the food aversion reflex we are studying [89].

Besides Mir-153, Mir-22 and Mir-29, several other microRNAs demonstrated differential expression after training selectively either in well-learned normal, or in learning-deficient DHT-treated animals (Table 2). So, in normal animals, there were changes in expression of Mir-7, -9, -76, -96, and -1985-S2_3p, while in learning-deficient DHT-treated animals, expression of other microRNAs (let-7, Mir-2, -92, -193, -277, -278, -279, -317, and -1992) changed. Some microRNAs were differentially expressed after training in both normal and DHT-treated animals, but with different dynamics. For example, Mir-33 expression decreased 5 h after training in normal animals and increased 2 h after training in DHT-treated animals. The other examples of microRNAs with treatment-dependent dynamics include Mir-133 and Mir-2722_3p. Thus, the above-mentioned microRNAs might play specific roles in the long-term plasticity of the food avoidance reflex in Helix. Several microRNAs (Mir-8, -71_3p, -87, -210, -216, and -1985-S1_5p) changed after training similarly in both normal and DHT-treated animals, which suggest involvement of these miRNAs in general processes associated with the training procedure, most likely a pain reaction. The above microRNAs are still seldom studied, or not studied at all in the context of neuroplasticity.

The expression of Mir-7, -9, -22, -33, -76, -96, -133, -1985-S2_3p, and -2722_3p changed in the CNS of normal *Helix* after learning. Mir-7 and Mir-9 play an important role in CNS development and pathologies [90,91]. Mir-9 is involved in synaptogenesis during early brain development, and an association has been found between these developmental events and adult cognition [91]. Mir-9 can regulate neurogenesis [92] and participates in the development of the olfactory system [93]. Mir-33 is involved in the regulation of GABAA receptor-mediated state-dependent fear encoding and retrieval [45]. Mir-96 is responsible for neurotoxins-induced cognitive dysfunction [94,95]. As a part of the miR-183/96/182 cluster, Mir-96 takes part in LTM formation [96]. In this case, as in our experiments, there is an increase in the expression after training. Reduced expression of the Mir-183/96/182 cluster is associated with memory impairment at the old age, and memory can be improved by overexpressing the cluster [15]. Mir-133 is associated with dopamine synthesis in vertebrates [97] and invertebrates [98], which may testify in favor of the conservation of not only their structure, but also their function. Mir-133b regulates production of tyrosine hydroxylase and the dopamine transporter [97], and is involved in the degeneration of the nigrostriatal dopaminergic system, which is the cause of Parkinson’s disease. Targets of Mir-133 include the receptor of epidermal growth factor [99]. Mir-133a is thought to be a potential molecular marker for the major depressive disorder [83].

In DHT-treated *Helix*, the expression of let-7, Mir-2, -92, -193, -277, -278, -279, -317, and -1992 changed after training. Dysregulation of let-7 family microRNAs is associated with the memory deficits in rat [16] and *Drosophila* [100], which indicates the functional conservation of this miRNA. Interestingly, let-7 is expressed in young nurse honeybees and is not expressed in old forager honeybees [76]. Mir-2 is among the ten most highly expressed miRNAs in the *Helix* CNS. Mir-2 is involved in neural development and maintenance in invertebrates [101]. The native targets of Mir-2-dependent gene silencing were suggested to represent putative neuroprotective modulators [102]. Mir-193 and Mir-277 are linked to memory formation in *Drosophila* [100,103]. Indeed, Mir-277 affects neurodegeneration in the *Drosophila* brain [104] and is involved in the regulation of its lifespan [105]. Let-7, -193, -277, and -279 miRNAs are consistently dysregulated in AD model in *Drosophila* [100]. Dysregulation of the above miRNAs in the CNS of DHT-treated *Helix* is most likely associated with dysfunction of the serotoninergic system and, accordingly, with impaired long-term memory and/or activation of adaptive mechanisms.

The expression of Mir-8, -71_3p, -87, -210, -216, and -1985-S1_5p similarly changed after training in normal and in DHT-treated animals. Mir-71 is absent in vertebrates, and its role in the functioning of the nervous system is poorly understood. However, it is interesting that Mir-71_5p is involved in the regulation of the chemotactic behavior and the lifespan of *C. elegans* [106] and is involved in the control of the formation of neuronal asymmetry [107]. In our experiments, Mir-71_3p is involved in food aversion learning, in which the conditioned stimulus is the initially attractive smell of food. Moreover, we have previously shown the presence of functional asymmetry in the *Helix* brain and its association with formation of food aversion [30]. Mir-87 is a critical regulator of dendrite regeneration in *Drosophila* [108]. Mir-210 directly regulates various human target genes associated with neuronal plasticity and neurodegenerative diseases [109]. One of the Mir-210 targets is the EphrinA3 mRNA [110,111]. Mir-210 is involved in visual pattern learning in honeybee, along with Mir-277, -278 and -279c [112]. The role of other above-mentioned microRNAs (miR-76, -216, -317, -1985, -1992, and -2722) in the mechanisms of neuroplasticity is currently unknown.

In summary, it can be noted that the expression of several microRNAs in the CNS of trained animals differs in comparison with naive animals. Remarkably, an unusually large number of MIR-10 family miRNAs are among them. It can be noted that 5 h after training, top differentially expressed representatives of the MIR-10 family demonstrated an increase in expression, which indicates a decrease in the expression of their potential targets. At the same time, the expression of most differentially expressed microRNAs of other families decreased, respectively, with a subsequent increase in the expression of their target genes. Interestingly, 2 h after training a different situation was observed. In addition, some miRNAs, which differentially expressed after training in DHT-untreated animals, did not change in DHT-treated animals, suggesting putative involvement of these miRNAs in the formation of food aversion. The expression of several microRNAs in DHT-treated and DHT-untreated animals was unidirectional, indicating their involvement in the general molecular mechanisms associated with training, most likely a pain reaction. The observed difference in the dynamics of expression of various representatives of the MIR-10 family may indicate the difference in their functional role in the studied reflex. Analysis of miRNA expression in the CNS in normal animals and in learning-deficient DHT-treated animals has shown that the change in the expression of a number of microRNAs is multidirectional (Table 2).

Our study demonstrates an important role of the serotoninergic system in the regulation of expression of miRNAs belonging to different families. Dysfunction of the serotoninergic system leads to changes in the metabolism of a number of miRNAs and, along with the epigenetic changes in the chromatin structure and modification of transcription factors, through regulation of genes might determine the deficiency of LTM formation in *Helix*. Our data support and extend the idea that the molecular machinery involved in long-term memory formation, including epigenetic tagging, is a conserved phenomenon during evolution.

Despite the fact that a large number of miRNAs expressed in the CNS have been identified, the mechanisms of regulation of their expression, as well as their targets, are poorly studied. The data obtained is often contradictory due to complexity of the structure and functioning of the brain, with varying functions of the miRNAs from the same family, and the opposite effects of different miRNAs on neuroplasticity [113]. Thus, our data on the presence of a wide range of highly conserved miRNAs in the *Helix* CNS, which play an important role in brain function, will make it possible successfully to use the *Helix* model to resolve the above problems.

## 4. Materials and Methods

### 4.1. Animals and Conditioned Reflex Formation

Experiments were carried out on adult (20–25 g) snails *Helix lucorum*, and did not require ethical approval. Snails had been in the active phase for not less than four weeks. All the animals were kept in terrariums. Prior to the isolation of the central nervous system, the animals were anaesthetized with ice-cold saline supplemented by injection of isotonic solution of MgCl_2_. Then the CNS (the circumoesophageal complex of ganglia) was excised quickly from the body.

Animals were trained to associate a piece of banana as the conditioned stimulus (CS) with an electric shock as the unconditioned stimulus (US). Similar procedure was used and described in [33]. Specifically, a piece of banana was placed at a distance of 1 cm from the head of a snail freely moving on a metal plate (serving as one of the stimulating electrodes). When the snail began to eat the banana, another stimulating electrode was placed manually on the snail’s head, and an electric shock (DC, 5 mA, 0.5 s) was applied. The training procedure consisted of 8 CS–US pairings applied at 15 min interval. Animals were deprived of food during the three days before the experiments. Naïve animals were used as a control group. Two or five hours after training, the snail’s CNS was excised and the total RNA sample was prepared.

The choice of time points was based on two considerations: the presence of two main waves of gene expression during formation of long-term memory [114], as well as data on the high stability of miRNAs [115]. It is known that the first wave is determined by activation of mRNA synthesis of immediate early genes, which include transcription factors, growth factors, and other regulatory proteins (the wave lasts up to two hours), and the second wave is determined by activation of mRNAs encoding other functional proteins (4–24 h). We chose the first time point of 2 h based on the works [8,116] and another point of 5 h (lies within the second wave). These points fall within the time interval used by Korneev and collegues [27], who studied miRNA dynamics in the CNS of *L. stagnalis* 1 and 6 h after one-trial conditioning, and are discussed in the article.

Separate groups of normal (DHT-untreated) and DHT-treated animals were tested for reflex retention three days after training by recording the time of approach and taking food (latent period).

### 4.2. Drugs and the Injection Procedure

To switch off the serotoninergic neurons, the animals were treated with selective neurotoxin 5,7-dihydroxytriptamine (Sigma) as described previously [29]. In short, 5,7-dihydroxytriptamine freshly dissolved in saline was injected at a dose of 20 μg/g of animal weight (0.2–0.3 mL) in two draughts at six day intervals. Injections were made in the cephalopedal sinus through an insensitive region of the body wall from the rear of the visceral sack. Learning experiments were carried out seven days after the last injection.

### 4.3. RNA Isolation and Sequencing

The total RNA was isolated from the circumesophageal ganglia with TRIzol™ Reagent (Invitrogen, Waltham, MA, USA). Small RNA fraction was isolated from the total RNA using the mirVana kit (Thermo Fisher Scientific, Waltham, MA, USA). The content and the quality of miRNA in the fraction of small RNA was determined on the BA2100 Bioanalyser with the Small RNA Kit (Agilent, Santa Clara, CA, USA). Experimental groups and the number of independent samples in sequencing analyses are as follows: naïve normal (5); naïve DHT-treated (5); 2 h post-training, normal (6); 2 h post-training, DHT-treated (6); 5 h post-training, normal (3); and 5 h post-training, DHT-treated (3). A total of 28 miRNA libraries were constructed using the TruSeq Small RNA Library Preparation Kit (Illimina, San Diego, CA, USA) according to the manufacturer’s protocol, 40 ng small RNA fraction as the starting amount, the T4 RNA ligase 2, deletion mutant (New England Biolabs, Ipswich, MA, USA) was used for the first ligation. Eleven PCR cycles were used for amplification, followed by AMPure XP (Agencourt, Brea, CA, USA) purification. Extraction of the fraction of libraries 145–160 bp containing the actual miRNA sequence was performed on a Caliper LabChipXT (PerkinElmer, Waltham, MA, USA). The quality and molarity of the final libraries obtained were determined on the BA2100 Bioanalyser using DNA High Sensitivity kit (Agilent, Santa Clara, CA, USA). Sequencing was performed on a NextSeq 550 (Illumina, San Diego, CA, USA) with the 36 bp reading length.

### 4.4. Bioinformatics Analysis

The sequencing results were quality filtered, followed by adapter removing and length selection (18–26 nt sequences were left for analysis). MiRNA candidates were predicted by the miRNA gFree program (https://bioinfo2.ugr.es/ceUGR/mirnagfree/, accessed on 20 October 2022). Only conserved miRNAs were selected from the prediction results for further research. The results of the sequencing were mapped on the *Lottia gigantea* and *Crassostrea gigas* conserved miRNAs (not predicted by the miRNAgFree for *H. lucorum*) obtained from the MirGeneDB database using Bowtie [117]. To remove the other RNAs, the prediction results were mapped by BLAST [118] on the RNA sequence of mollusks (except for the miRNA sequences) from the Rfam (13.0), RefSeq (Release 86), and GeneBank (Release 223).

### 4.5. cDNA Synthesis and Real-Time RT-PCR

The cDNA was obtained by a reverse transcription reaction with the specific stem-loop primers for each miRNA using RevertAid reverse transcriptase (Termo Fisher Scientific, Waltham, MA, USA), according to the manufacturer’s protocol. The resulting cDNA was diluted five times. The levels of the miRNA transcription were evaluated by the TaqMan real-time PCR on a CFX96 Real-Time PCR Detection System (Bio-Rad Laboratories, Hercules, CA, USA). The significance of the results was accepted at *p* < 0.05.

## Figures and Tables

**Figure 1 ijms-24-00301-f001:**
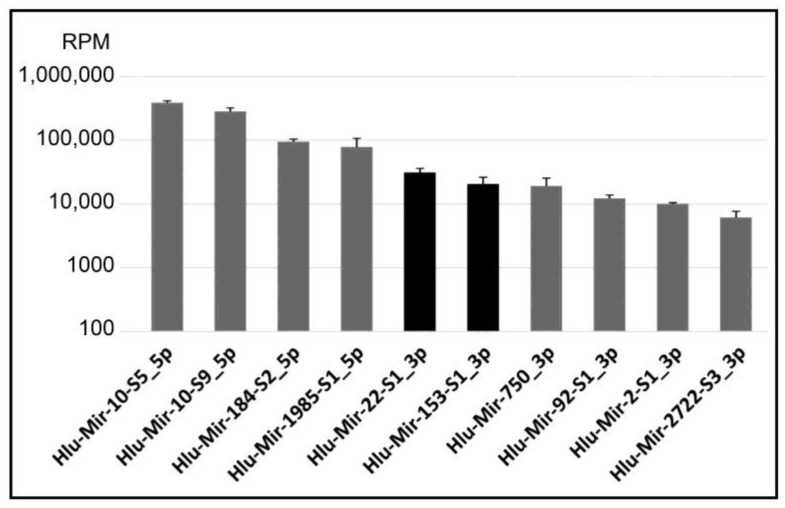
The expression levels of the top 10 most abundant miRNAs in *H. lucorum* CNS. The data shown are mean ± SD normalized in reads per million (RPM). The expression levels of remaining miRNAs are presented in Appendix A.

**Figure 2 ijms-24-00301-f002:**
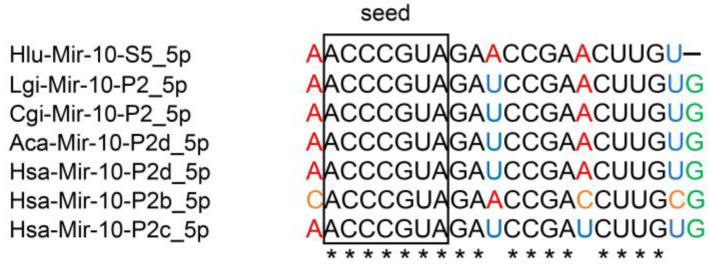
MIR-10 family miRNAs alignment. The seed plot is marked with a rectangle. The differences in the sequences are highlighted by color. Species designations: has-*H. sapiens*, Hlu-*H. lucorum*, Lgi-*L. gigantea*, Cgi-*C. gigas*, Aca-*A. californica*. Alignments for some other miRNAs of the MIR-10 family are presented in Appendix A. * indicates top conserve nucleotides.

**Figure 3 ijms-24-00301-f003:**
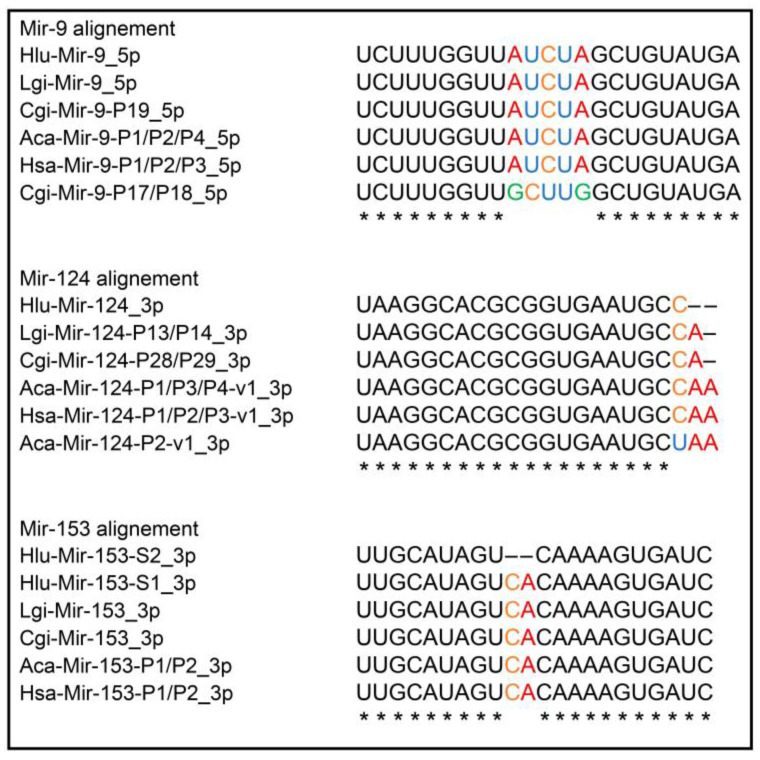
Alignment of human’s and mollusks’ mature miRNA sequences belonging to the families MIR-9, MIR-124, and MIR-153. Differences in the sequences are highlighted by color. Species designations: Hsa-*H. sapiens*, Hlu-*H. lucorum*, Lgi-*L. gigantea*, Cgi-*C. gigas*, Aca-*A. californica*. For some other miRNAs, comparative analysis data are presented in Appendix A. * indicates top conserve nucleotides.

**Figure 4 ijms-24-00301-f004:**
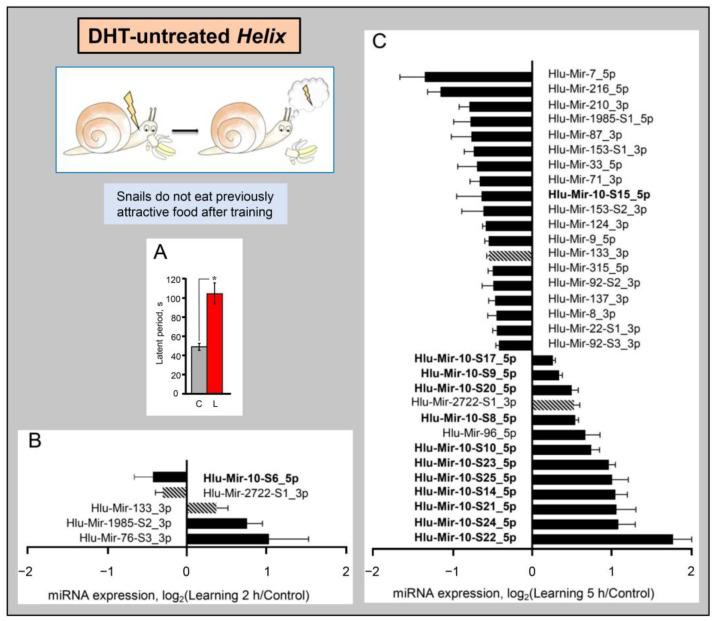
Learning regulates miRNAs expression in *Helix* CNS. (**A**) The formation of the food-avoidance conditioning in Helix. Testing after 3 days shows significant learning effects. The ordinate is the latent period of the consummatory responses of animals. Designations: C-animals before training (*n* = 12), L-trained animals (*n* = 12). * *p* < 0.05. (**B**,**C**)-Relative expression levels of miRNAs 2 h (**B**) and 5 h (**C**) after learning were assessed using next-generation sequencing normalizing first to the total number of miRNA reads in each sample and then to the mean expression value in control groups of naive animals. The hatched columns indicate miRNAs which show differential expression both 2 and 5 h after learning. Statistical significance was assessed using the U-test (*p* < 0.05). The error bars represent the standard error of the mean. The number of samples in experimental groups is as follows: Control, *n* = 10; 2 h post-training, *n* = 6; 5 h post-training, *n* = 3.

**Figure 5 ijms-24-00301-f005:**
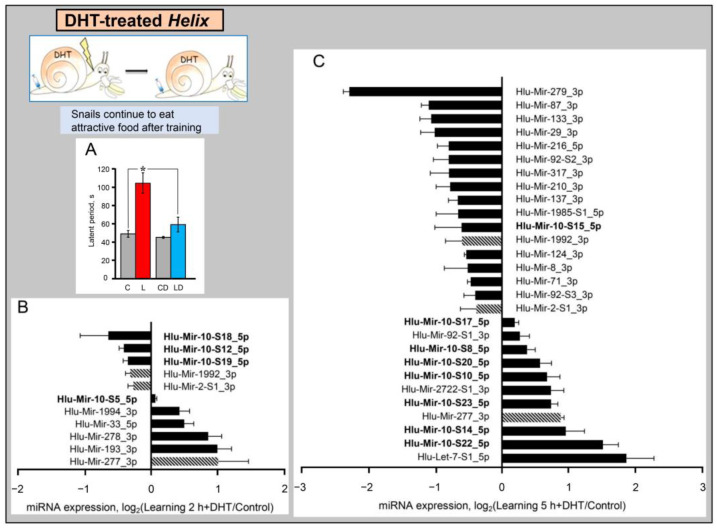
Training regulates miRNA expression in the CNS of DHT-treated *Helix*. (**A**) The pretreatment by the neurotoxin 5,7-DHT interferes formation of the food aversion reflex in Helix. The ordinate is the latent period of the consummatory responses of animals. Designations: C and CD -animals before training, normal (*n* = 12), and DHT-treated (*n* = 8), correspondingly; L and LD -trained animals, normal (*n* = 12) and DHT-treated (*n* = 8), correspondingly. Repeated measures ANOVA [Treatment (Normal, DHT-treated) x Training (Before, 3 days after)] with repeated measures in the factor “Training” showed a significant interaction between the factors, F (1, 18) = 5.0, *p* = 0.039. * *p* < 0.05. (**B**,**C**) -Relative RNA expression 2 h (**B**) and 5 h (**C**) after training was assessed using next-generation sequencing normalizing first to the total number of miRNA reads in each sample and then to the mean expression value in control groups of naive animals. The hatched columns indicate miRNAs, which show similar dynamics 2 and 5 h after training. Statistical significance was assessed using the U-test (*p* < 0.05). The error bars represent the standard error of the mean. The number of samples in experimental groups is as follows: Control, *n* = 10; 2 h post-training, DHT-treated, *n* = 6; 5 h post-training, DHT-treated, *n* = 3.

**Figure 6 ijms-24-00301-f006:**
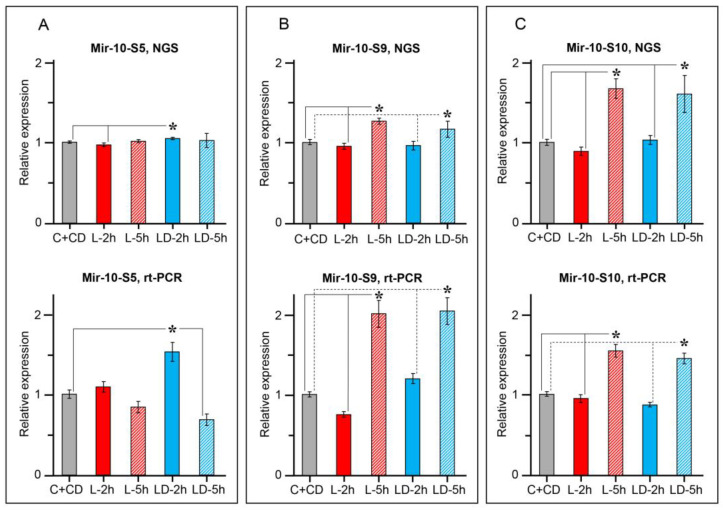
MicroRNAs of the MIR-10 superfamily regulated in the CNS of *H. lucorum* after training. (**A**) Mir-10-S5, (**B**) Mir-10-S9, (**C**) Mir-10-S10. The results of next-generation sequencing (NGS) were confirmed by real time PCR. Experimental groups and the number of independent samples in sequencing/PCR analyses are as follows: naïve normal (5/7); naïve DHT-treated (5/6); 2 h post-training, normal (6/7); 2 h post-training, DHT-treated (6/8); 5 h post-training, normal (3/9); 5 h post-training, DHT-treated (3/6). C+CD -general control groups of naïve animals, normal, and DHT treated; L-2 h/5 h and LD-2 h/5 h—trained animals, normal (L) or DHT treated (LD), 2 or 5 h after training. Data are presented as “mean ± SEM”. * *p* < 0.05.

**Table 1 ijms-24-00301-t001:** MiRNA superfamily MIR-10 expressed in the CNS of *H. lucorum* and the most similar human homologues.

Subfamily	Seed Region	*Helix* miRNA	Human Homologue
**MIR-10-P1** **(mir-10)**	**ACCCUGU**	**from hlu-mir-10-S1 to hlu-mir-10-S3**	**hsa-mir-10a** **[Gene ID: 406902]**
**MIR-10-P2** **(mir-99/100)**	**ACCCGUA**	**from hlu-mir-10-S4 to hlu-mir-10-S7**	**hsa-mir-100** **[Gene ID: 406892]**
**MIR-10-P3** **(mir-125)**	**CCCUGAG**	**from hlu-mir-10-S8 to hlu-mir-10-S26**	**hsa-mir-125b** **[Gene ID: 406911]**

**Table 2 ijms-24-00301-t002:** Food aversion learning is associated with time-dependent changes in microRNA profiles in the CNS of *H. lucorum* compared to naive animals. The damage of serotoninergic transmission by DHT impairs the learning and shifts dynamics of several microRNAs 2 h and 5 h after training. **⇩**-downregulation, **↑**-upregulation.

MicroRNA dynamics 2 h after training					
MicroRNA	Normal animals	DHT treated	MicroRNA	Normal animals	DHT treated
*Hlu-Mir-2-S1_3p*		**⇩**	*Hlu-Mir-76-S3_3p*	**↑**	
**Hlu-Mir-10-S5_5p**		**↑**	*Hlu-Mir-133_3p*	**↑**	
**Hlu-Mir-10-S6_5p**	**⇩**		*Hlu-Mir-193_3p*		**↑**
**Hlu-Mir-10-S12_5p**		**⇩**	*Hlu-Mir-277_3p*		**↑**
**Hlu-Mir-10-S18_5p**		**⇩**	*Hlu-Mir-278_3p*		**↑**
**Hlu-Mir-10-S19_5p**		**⇩**	*Hlu-Mir-1985-S2_3p*	**↑**	
*Hlu-Mir-33_5p*		**↑**	*Hlu-Mir-2722-S1_3p*	**⇩**	
MicroRNA dynamics 5 h after training					
MicroRNA	Normal animals	DHT treated	MicroRNA	Normal animals	DHT treated
*Hlu-Let-7-S1_5p*		**↑**	*Hlu-Mir-71_3p*	**⇩**	**⇩**
*Hlu-Mir-2-S1_3p*		**⇩**	*Hlu-Mir-87_3p*	**⇩**	**⇩**
*Hlu-Mir-7_5p*	**⇩**		*Hlu-Mir-92-S1_3p*		**↑**
*Hlu-Mir-8_3p*	**⇩**	**⇩**	*Hlu-Mir-92-S2_3p*	**⇩**	**⇩**
*Hlu-Mir-9_5p*	**⇩**		*Hlu-Mir-92-S3_3p*	**⇩**	**⇩**
**Hlu-Mir-10-S8_5p**	**↑**	**↑**	*Hlu-Mir-96_5p*	**↑**	
**Hlu-Mir-10-S9_5p**	**↑**	**↑**	*Hlu-Mir-124_3p*	**⇩**	**⇩**
**Hlu-Mir-10-S10_5p**	**↑**	**↑**	*Hlu-Mir-133_3p*	**⇩**	**⇩**
**Hlu-Mir-10-S14_5p**	**↑**	**↑**	*Hlu-Mir-137_3p*	**⇩**	**⇩**
**Hlu-Mir-10-S15_5p**	**⇩**	**⇩**	*Hlu-Mir-153-S1_3p*	**⇩**	
**Hlu-Mir-10-S17_5p**	**↑**	**↑**	*Hlu-Mir-153-S2_3p*	**⇩**	
**Hlu-Mir-10-S20_5p**	**↑**	**↑**	*Hlu-Mir-210_3p*	**⇩**	**⇩**
**Hlu-Mir-10-S21_5p**	**↑**		*Hlu-Mir-216_5p*	**⇩**	**⇩**
**Hlu-Mir-10-S22_5p**	**↑**	**↑**	*Hlu-Mir-277_3p*		**↑**
**Hlu-Mir-10-S23_5p**	**↑**	**↑**	*Hlu-Mir-279_3p*		**⇩**
**Hlu-Mir-10-S24_5p**	**↑**		*Hlu-Mir-317_3p*		**⇩**
**Hlu-Mir-10-S25_5p**	**↑**		*Hlu-Mir-1985-S1_5p*	**⇩**	**⇩**
*Hlu-Mir-22-S1_3p*	**⇩**		*Hlu-Mir-1992_3p*		**⇩**
*Hlu-Mir-29_3p*		**⇩**	*Hlu-Mir-2722-S1_3p*	**↑**	**↑**
*Hlu-Mir-33_5p*	**⇩**				

## Data Availability

Data reported in this study were collected and stored by L.N.G.

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
