# Peer review of "The Expression of miRNAs Involved in Long-Term Memory Formation in the CNS of the Mollusk Helix lucorum"

_ijms, 2022, doi:10.3390/ijms24010301_

Round 1
Reviewer 1 Report
This is the first comprehensive identification of miRNAs in the terrestrial snail Helix lucorum, a highly useful molluscan model organism for the analysis of cellular and molecular mechanisms of LTM formation after food-avoidance conditioning. The authors used a combination of behavioral, RNA sequencing and bioinformatics methods to identify 95 different miRNAs, compare them to miRNAs identified in other species, and show the involvement of MIR-10 family miRNAs in different stages of memory consolidation in normal and learning-deficient animals. Overall, I rate this a potentially excellent study but I have a number of major and minor comments regarding a number of issues that need to be addressed by the authors.
Major
1. The authors do not show any data on the outcome of the classical conditioning experiments. The expectation with this type of experiment is that a control and a classically conditioned group of animals is tested 24 hours after training with other groups sacrificed at different times after training to provide the samples for the molecular analysis.
2. The authors do not provide any explanation, either in the Introduction or Results, for the use of the 2 h and 5 h post-training time points for their comparisons of miRNA levels; they must do this in the revised manuscript.
3. In the experiments presented in Figs. 4 and 5. the n numbers in the 5-hour group (3) appear to be too low for meaningful statistical comparisons with the control data. Can the authors justify their conclusions based on such a small sample?
Minor
1. The text will require careful proofreading and extensive correction of formatting issues as well as the use of English in the text of the manuscript and also in the figures. With regard to the formatting issues, for example, there are numerous examples of a lack of spaces between words, which made the reading of the text difficult.
2. Consistent italisation of all species names, whether abbreviated or not (e.g., Helix, Helix lucorum, A. californica, Aplysia californica) is required throughout the text.
3. Remove the author instruction text from the beginning of the Results section.
4. The text on most of the figures appears to be blurred, please use higher-resolution originals.
Author Response
Please, see the attached file

Reviewer 2 Report
There is lacking experimental evidence to prove that miRNAs are involved in physiological functions such as learning and memory. There is lacking phylogenetic analysis for the evolution of miRNA for confirming the conserved function. Also, the author should analyze the downstream transcriptionally regulated genes for proving their conserved functions. Importantly, there are lots of language mistakes in the manuscript and making it really hard to read through the manuscript (lines 74, 145, 225, 226). In Figure 1, there was no x-label.
Author Response
Please, see the attachement

Round 2
Reviewer 1 Report
The manuscript is much improved but there are a number of minor issues that the authors may want to address to improve it even further. My comments are as follows:
* Fig. 4 label: 'Snails do not eat', not 'does not eat'.
* Fig. 5 label: italicise 'Helix'
* Make layout of Fig. 5 same as Fig. 4.
* Use Arial font consistently in all figure labels (like in the label
of Fig. 6).
* English still needs some improvements.
I recommend acceptance with minor corrections.
Author Response
Dear Reviewer,
Thank you for the revision of our manuscript, critical comments and constructive suggestions.
We took into account all the comments and made the necessary corrections to the article.

Reviewer 2 Report
NA
Author Response
Dear Reviewer,
Thank you for the revision of our manuscript.
We took into account the comments and corrected English translation.